# Multilingual Safety Alignment via Representation-Space Separability

Dan Shi [* 1]   Zhuowen Han [* 1]   Deyi Xiong [1]

## Abstract

Large language models (LLMs) have been globally adopted in various scenarios, making robust multilingual safety alignment a prerequisite for their reliable deployment across diverse languages. Despite recent advances, LLMs exhibit a substantial safety gap between high- and low-resource languages: models that can consistently refuse harmful requests in high-resource languages often fail to do so in low-resource languages. In this work, we reveal that such safety failures stem from insufficient representation-space separability between harmful and harmless prompts in low-resource languages. Through geometric analyses, we find that, compared to English, harmful prompts are significantly less separated from the manifold of harmless prompts, and that the resulting cross-lingual spatial margin gap is strongly correlated with attack success rates. Capitalizing on these insights, we propose Multilingual **S**patial **M**argin Gap-based **O**ptimization (SMO), a novel training strategy that exploits the well-aligned safety geometry of a dominant language (e.g., English) to enhance safety alignment in other languages. SMO explicitly leverages the spatial margin gap between English and target languages as an example-wise supervision signal, enabling effective cross-lingual transfer of safety capabilities while preserving the dominant language's original performance. Experiments conducted on LLaMA-3.1-8B-Instruct and Qwen2.5-7B-Instruct demonstrate that SMO is capable of substantially reducing attack success rates in low-resource languages to near zero, often reaching zero, while maintaining strong general multilingual performance. Warning: This paper contains content that may be harmful.

---

[*]Equal contribution [1]TJUNLP Lab, School of Computer Science and Technology, Tianjin University, Tianjin, China. Correspondence to: Deyi Xiong <dyxiong@tju.edu.cn>.

*Proceedings of the 43rd International Conference on Machine Learning*, Seoul, South Korea. PMLR 306, 2026. Copyright 2026 by the author(s).

## 1. Introduction

Large language models (LLMs) have seen widespread deployment in global applications (Touvron et al., 2023; Grattafiori et al., 2024; Jaech et al., 2024; Team et al., 2025; Guo et al., 2025; 2023), making robust multilingual safety behavior a critical requirement (Zhu et al., 2024; Shi et al., 2024b). Despite significant progress in safety alignment for high-resource languages (Shen et al., 2023), especially English, recent studies consistently report a pronounced multilingual safety gap: models that reliably refuse harmful requests in English often fail to do so in low-resource languages (Deng et al.; Yong et al., 2023; Xu et al., 2024; Wang et al., 2024; Shen et al., 2024).

Recent mechanistic analyses offer an initial explanation for this phenomenon. In particular, Wang et al. (2025) suggest that safety-aligned models share a universal refusal mechanism across languages, which can be characterized by a consistent direction in representation space associated with refusal behavior. This observation implies that multilingual safety failures may not stem from the absence of refusal capabilities in low-resource languages. Rather, it raises the hypothesis that failures arise when models do not reliably recognize that a given prompt warrants refusal in those languages. However, this hypothesis has not yet been systematically validated or quantified through direct empirical analysis.

To investigate this question, we first conduct an in-depth analysis on the representation-space geometry of harmful and harmless prompts across languages on widely-used LLMs, including LLaMA-3.1 (Grattafiori et al., 2024) and Qwen2.5 (Yang et al., 2024), which have undergone sufficient safety alignment for their dominant language (typically English). Using PCA and distance-based measurements, we uncover a consistent pattern: harmful prompts in low-resource languages are substantially less separated from harmless prompts than in English, as demonstrated in Figure 1 and 2. More importantly, this geometric deficiency is strongly predictive of downstream safety outcomes. We find that the gap between English and target-language spatial margins, defined as the distance between harmful prompts and harmless prompts, correlates closely with attack success rates across languages. As shown in Table 1 in §2.4, languages exhibiting larger separability gaps consistently show

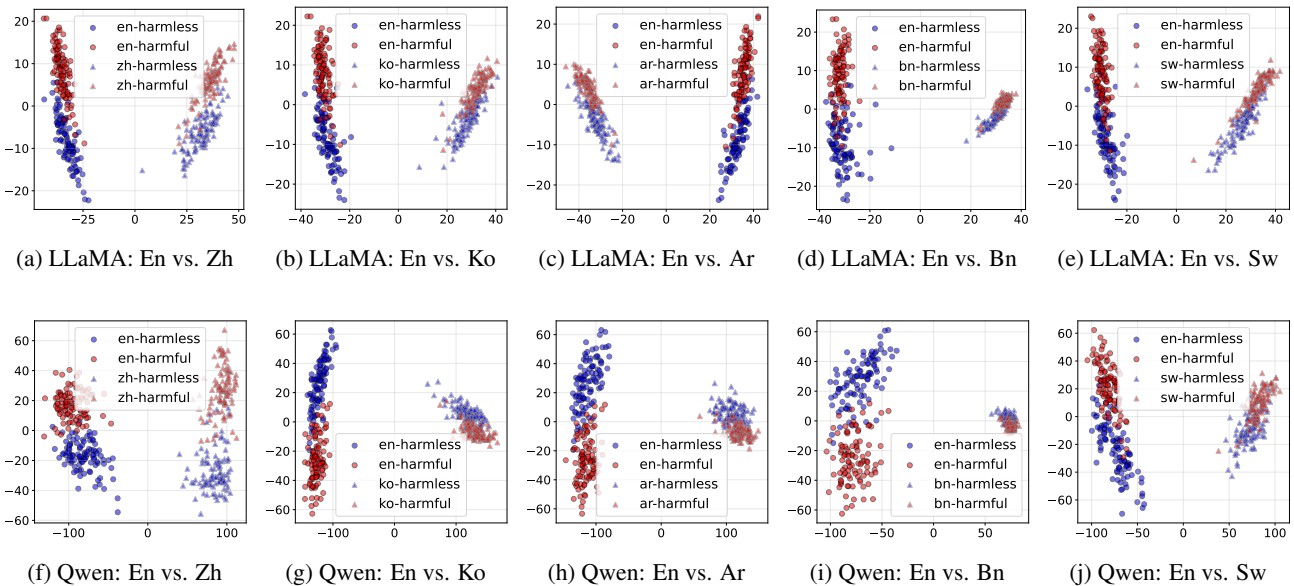

Figure 1. PCA visualizations of multilingual harmful vs. harmless prompt representations from the final layer. Top row: `Llama3.1-8B-Instruct`. Bottom row: `Qwen2.5-7B-Instruct`.

higher vulnerability to harmful prompts, whereas smaller gaps correspond to substantially lower attack success. Together, these results confirm that multilingual safety failures arise not from missing refusal capabilities, but from insufficient recognition of harmful prompts in representation space. In other words, *the failure lies not in how the model refuses, but in whether it recognizes a prompt as warranting refusal*.

This observation reframes multilingual safety alignment as a problem of representation-space separability. If a model internally recognizes a prompt as harmful, its representation should lie sufficiently far from the manifold of harmless prompts, enabling downstream refusal behavior. Conversely, when harmful prompts remain close to the harmless manifold, refusal signals are weak or ambiguous, particularly in low-resource languages. Despite its conceptual importance, this geometric perspective has not been explicitly operationalized in existing multilingual safety training methods.

In this work, we propose Multilingual **S**patial **M**argin Gap-based **O**ptimization (SMO), a representation-space approach to multilingual safety alignment that directly targets this failure mode. As the core component of SMO, we introduce **spatial margin**, defined as the distance between the hidden representation of the prompt of a training sample and the centroid of harmless prompt representations. This margin serves as a quantitative measure of whether the model internally distinguishes a prompt from the harmless manifold. We compute spatial margins separately for English and a target language, using the same underlying model representations.

The key insight underlying SMO is that the gap between English and target-language spatial margins provides an example-wise signal of safety uncertainty. If a harmful prompt is well separated from the harmless manifold in English but not in the target language, the model likely fails to recognize that the prompt should trigger refusal in that language. We leverage this margin gap to modulate optimization strength during training: examples with insufficient separability in the target language receive stronger updates, while those already well separated require less intervention. To ensure stability and preserve behavior in English, we decouple this margin signal from gradient flow and incorporate a retain loss that anchors English representations.

This formulation has several desirable properties. First, it aligns directly with the universality of refusal mechanisms: rather than learning new refusal behaviors, SMO amplifies the model's ability to recognize harmful prompts across languages. Second, by operating entirely in representation space, it provides a geometric interpretation of multilingual safety failures and their correction. Third, the example-wise modulation of optimization strength enables targeted improvement without degrading general capabilities.

We evaluate our method on three instruction-tuned model families, LLaMA-3.1-8B-Instruct and Qwen2.5-7B-Instruct, across multiple low-resource languages. Our experiments demonstrate substantial reductions in attack success rates, often eliminating successful attacks entirely. Meanwhile, SMO preserves general-purpose multilingual capabilities. Further in-depth analyses corroborate the effectiveness of the cross-lingual spatial-margin gap as a more reliable and

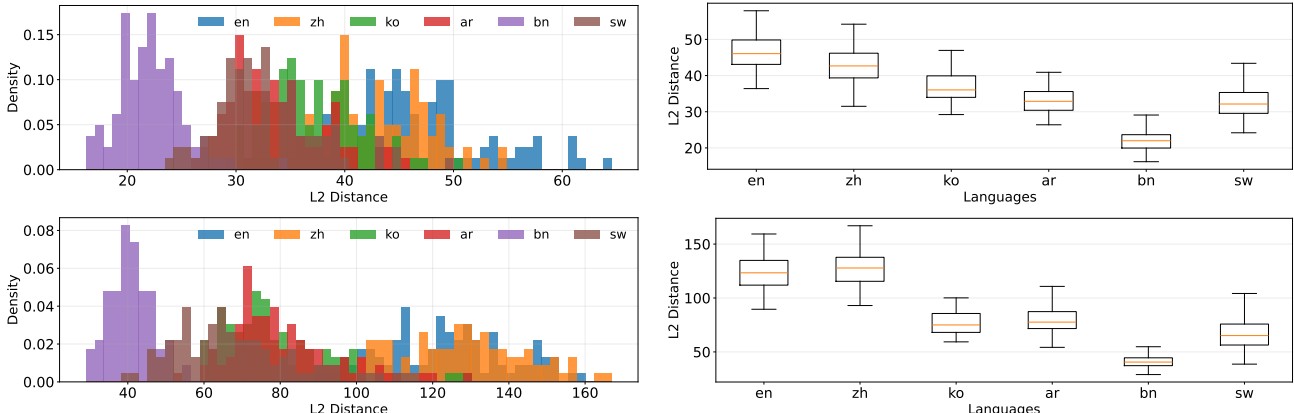

Figure 2. Distribution of L2 distances between harmful and harmless prompt representations across all languages on `Llama3.1-8B-Instruct` (top row) and `Qwen2.5-7B-Instruct` (bottom row). For each model, the left panel shows kernel-smoothed histograms of the distances for the six languages. The right panel summarizes the same statistics using box plots. English exhibits consistently larger distances, whereas non-English languages show progressively smaller distances, indicating reduced representational separation in low-resource languages.

scalable supervision signal for multilingual safety alignment.

In summary, our contributions are threefold:

- We frame multilingual safety failures as a problem of representation-space separability, grounded in the universality of refusal mechanisms.

- We propose SMO, which employs the spatial margin gap difference between the dominant language and target languages as a high-quality optimization supervision signal, enabling effective multilingual safety alignment.

- We demonstrate consistent and substantial safety improvements across multiple model families and languages without sacrificing general capabilities.

## 2. Preliminaries

In this section, we present a geometric analysis of multilingual safety behavior in representation space. Our goal is to characterize how harmful and harmless prompts are separated across languages, and to examine how this separation relates to multilingual safety performance.

### 2.1. Setup

**Models** We select one English-centric LLMs: `LLaMA-3.1-8B-Instruct` (Grattafiori et al., 2024) and one bilingual LLMs: `Qwen2.5-7B-Instruct` (Yang et al., 2024), to demonstrate the harmful and harmless representation space gap on safety issues across different languages.

**Dataset** We begin with the English dataset used by (Arditi et al., 2024), which contains two subsets: a harmful prompt set $D_{harmful}$ and a harmless prompt set $D_{harmless}$. The harmful prompt set $D_{harmful}$ consists of malicious instructions collected from ADVBENCH (Zou et al., 2023b), MALICIOUSINSTRUCT (Huang et al., 2023), and TDC2023 (Mazeika et al., 2023). The harmless prompt set $D_{harmless}$ contains prompts from Alpaca (Taori et al., 2023).

**Languages** To study multilingual behavior, we select six languages spanning different language families and resource levels. High-resource languages include English (en) and Chinese (zh); medium-resource languages include Korean (ko) and Arabic (ar); low-resource languages include Bengali (bn) and Swahili (sw). To create a multilingual version of the dataset, following Wang et al. (2025), we translate the original English prompts into the other 5 languages using Google Translate[1].

For each prompt, we extract its hidden representation from the final transformer layer of the model. Specifically, given a prompt consisting of multiple tokens, we compute the prompt-level representation by averaging the hidden states of all tokens in the final layer. This representation is used consistently across all analyses in this section.

### 2.2. PCA Analysis of Representation Geometry

We first visualize the representation-space geometry of harmful and harmless prompts using principal component analysis (PCA). For each language, we project the prompt-level representations of harmful and harmless inputs onto

---

[1]https://github.com/nidhaloff/deep-translator

the first two principal components, computed jointly across languages.

Figure 1 presents PCA visualizations for English paired with each target language. A clear and consistent pattern emerges across both model families. In English, harmful and harmless prompts form two well-separated clusters, indicating strong representation-space separability. In contrast, for non-English languages, this separability progressively degrades as language resource availability decreases. In particular, for low-resource languages such as Bengali and Swahili, the representations of harmful and harmless prompts are substantially more entangled, with significant overlap between the two groups.

### 2.3. Distance-Based Separability

While PCA visualizations provide an intuitive view of representation geometry, we further quantify harmful–harmless separability using a direct distance-based measure. For all harmful and harmless prompts in the dataset, we obtain all the representations. Then we compute the $\ell_2$ distances over all harmful–harmless combinations. The resulting distance distributions characterize how well harmful prompts are separated from harmless prompts in representation space.

Figure 2 shows the distance distributions across languages using both histograms and box plots. A clear cross-lingual trend emerges. In English, harmful–harmless distances are consistently larger, indicating strong representational separation. In contrast, distances shrink progressively for lower-resource languages.

### 2.4. The Correlation between Separability and Attack Success Rates

We further quantitatively investigate the relationship between representation-space separability and downstream safety performance. To enable a direct cross-lingual comparison, we define a *distance gap* that measures how much less separated harmful and harmless prompts are in a target language relative to English.

Specifically, for each language $\ell$, we compute the average harmful-harmless distance

$$\overline{D}_\ell = \mathbb{E}_{(\mathbf{h},\mathbf{u})\sim\ell}\left[\|\mathbf{h}-\mathbf{u}\|_2\right],$$

where $(\mathbf{h}, \mathbf{u})$ denotes a harmful-harmless prompt pair in language $\ell$. We then define the *distance gap* between English and a target language $\ell$ as

$$\Delta D_\ell = \overline{D}_{\text{en}} - \overline{D}_\ell.$$

In parallel, we quantify the gap in safety performance using the attack success rate (ASR) metric. Let $\text{ASR}_\ell$ denote the attack success rate in language $\ell$. The corresponding *ASR*

Table 1. Representation-space distance gaps and corresponding ASR gaps between English and target languages. Across model families, larger distance gaps are almost consistently associated with larger safety performance gaps.

|  | Zh | Ko | Ar | Bn | Sw |
|---|---|---|---|---|---|
| **Llama3.1-8B-Instruct** | | | | | |
| Distance GAP | 4.61 | 10.34 | 13.61 | 25.13 | 14.41 |
| ASR GAP | -5.72 | -37.78 | -4.03 | -34.92 | -23.18 |
| **Qwen2.5-7B-Instruct** | | | | | |
| Distance GAP | -3.47 | 45.53 | 43.00 | 82.86 | 55.53 |
| ASR GAP | -6.98 | -12.07 | -1.59 | -17.78 | -12.06 |

*gap* with English is defined as

$$\Delta\text{ASR}_\ell = \text{ASR}_{\text{en}} - \text{ASR}_\ell.$$

The results is demonstrated in Table 1. Across both LLMs, we observe a clear and consistent trend: languages with larger distance gaps relative to English tend to exhibit substantially larger safety performance gaps. This relationship holds across model families and evaluation settings, indicating that representation-space separability is a reliable predictor of multilingual safety performance.

The foregoing analyses indicate that, in low-resource languages, harmful prompts tend to remain closer to harmless prompts in the representation space, rendering them more difficult for the model to distinguish reliably. More importantly, these results establish a direct link between representation-space separability and failures in multilingual safety. Taken together, the analyses support the conclusion that multilingual safety failures stem from insufficient separation between harmful and harmless prompts in the representation space, with this issue being particularly pronounced for low-resource languages. This observation directly motivates our representation-space–based approach to multilingual safety alignment, which we introduce in the following section.

## 3. Method

Based on the above insights, we propose a novel method for multilingual safety alignment called Multilingual **S**patial **M**argin Gap-based **O**ptimization (SMO), which leverages the well-aligned separation between harmful and harmless prompts in a dominant language (typically English) as an anchor, and explicitly aligns the representation-space separability of target languages to it. Figure 3 illustrates the SMO framework.

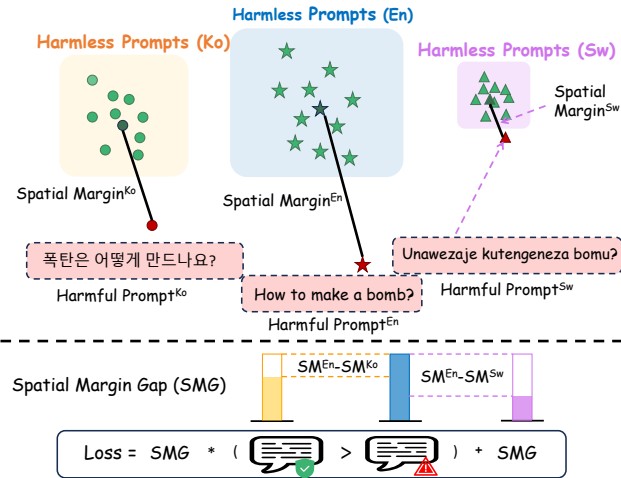

Figure 3. Overview of SMO. Given a harmful instruction in a dominant language (e.g., English) and a target language, SMO computes language-specific spatial margins as the distance between each harmful prompt representation and the corresponding harmless centroid. The resulting cross-lingual spatial margin gap quantifies safety discrimination in the target language. This gap is then used both as an explicit margin alignment loss and as an example-wise modulation signal for targeted safety alignment.

## 3.1. Harmless Representation Centroids

We begin by computing harmless representation centroids, which serve as reference points for measuring harmful-harmless separability. Let $\mathcal{D}^\ell_{\text{harmless}}$ denote the set of harmless prompts of language $\ell$. For each prompt, we obtain its hidden representation by mean-pooling all token representations from the final transformer layer. We then compute a language-specific harmless centroid as

$$c^\ell = \frac{1}{|\mathcal{D}^\ell_{\text{harmless}}|} \sum_{x \in \mathcal{D}^\ell_{\text{harmless}}} h(x), \qquad (1)$$

where $h(x)$ denotes the hidden representation of prompt $x$. These centroids characterize the harmless manifolds in representation space for each language and provide a stable geometric reference for evaluating the separation of harmful prompts.

## 3.2. Spatial Margins and Margin Gap

Our training data consists of paired bilingual preference examples derived from a dominant language $d$ and a target language $t$. Each training instance consists of two triplets: $(x^t, y^t_w, y^t_l)$ and $(x^d, y^d_w, y^d_l)$, where $x$ denotes a harmful instruction, $y_w$ is the preferred (refusal) response, and $y_l$ is the rejected (unsafe) response. The target-language triplet is obtained via translation of the dominant-language instruction, ensuring semantic equivalence across languages.

For a harmful instruction $x$ in language $\ell$, we define its **spatial margin** (SM) as the $\ell_2$ distance between its representation and the harmless centroid of that language:

$$\text{SM}^\ell(x) = \left\| h(x) - c^\ell \right\|_2. \qquad (2)$$

For each paired training example, we compute the spatial margins for the dominant language $d$ and the target language $t$:

$$\text{SM}^d = \text{SM}^d(x^d), \quad \text{SM}^t = \text{SM}^t(x^t). \qquad (3)$$

To transfer the safety capability from the dominant language to the target language, we optimize the **spatial margin gap** between them, defined as:

$$\mathcal{L}_{\text{margin}} = \max\left(0, \ \text{SM}^d - \text{SM}^t\right). \qquad (4)$$

This loss penalizes cases where a harmful prompt is well separated from the harmless manifold in the dominant language but insufficiently separated in the target language. Conversely, target-language prompts that already exhibit adequate separation incur no penalty.

## 3.3. Optimization with SimPO

We integrate the margin loss into preference-based safety training using SimPO (Meng et al., 2024). SimPO is an improved variant of DPO (Rafailov et al., 2023), which directly optimizes the preferences between refusal and unsafe responses without requiring a reference model:

$$\mathcal{L}_{\text{SimPO}}(\theta) = -\mathbb{E}\Big[\log\sigma\big(\frac{\beta}{|y^t_w|}\log\pi_\theta(y^t_w \mid x^t) \\ -\frac{\beta}{|y^t_l|}\log\pi_\theta(y^t_l \mid x^t) - \gamma\big)\Big]. \qquad (5)$$

In the context of safety alignment, SimPO provides a more precise reward measure (Zhao et al., 2025). The incorporation of length normalization mitigates reward errors caused by length bias (Singhal et al.; Park et al., 2024): unsafe responses often contain detailed harmful content and are therefore longer, whereas safe responses typically exhibit concise refusal patterns.

Beyond serving as an auxiliary objective, the spatial margin gap also provides an example-wise signal of safety recognition. If $\text{SM}^t \ll \text{SM}^d$, the model exhibits weak harmful-harmless separability in the target language, indicating that stronger optimization is required for this example. To capture this intuition, we also detach the spatial margin gap from gradient flow (denoted as SMG) and use it to modulate the optimization strength of each example in SimPO:

$$\mathcal{L} = \text{SMG} \cdot \mathcal{L}_{\text{SimPO}} + \lambda\mathcal{L}_{\text{margin}}, \qquad (6)$$

Table 2. Main results on MultiJail and AdvBench-X datasets. Best results are **bolded**, and second-best are underlined.

| Model | MultiJail | | | | | | | AdvBench-X | | | | | | |
|---|---|---|---|---|---|---|---|---|---|---|---|---|---|---|
| | **En** | **Zh** | **Ko** | **Ar** | **Bn** | **Sw** | **AVG.** | **En** | **Zh** | **Ko** | **Ar** | **Bn** | **Sw** | **AVG.** |
| | *Base Model: LLaMA-3.1-8B-Instruct* | | | | | | | | | | | | | |
| **Base** | 10.48 | 19.05 | 46.98 | 14.92 | 59.37 | 77.46 | 38.04 | 4.42 | 3.27 | 25.00 | 7.31 | 60.58 | 68.85 | 28.24 |
| SFT | 8.89 | 16.63 | 43.49 | 14.29 | 61.90 | 76.19 | 36.93 | 4.42 | 2.69 | 25.19 | 7.50 | 60.19 | 68.27 | 28.04 |
| DPO | 2.22 | 1.59 | 10.79 | 0.63 | 32.70 | 48.25 | 16.03 | **0.00** | 0.58 | 7.50 | 0.58 | 14.42 | 30.38 | 8.91 |
| ORPO | 5.08 | 4.44 | 20.00 | 4.13 | 31.75 | 38.73 | 17.35 | 0.19 | 0.96 | 4.81 | 1.35 | 16.92 | 18.65 | 7.15 |
| KTO | 3.17 | 1.27 | 12.38 | 0.95 | 21.27 | 27.94 | 11.16 | **0.00** | 0.38 | 2.31 | 0.19 | 9.42 | 13.46 | 4.29 |
| SimPO | 1.90 | 0.63 | 4.44 | **0.00** | 13.97 | 14.29 | 5.87 | **0.00** | 0.19 | 2.31 | **0.00** | 9.04 | 2.69 | 2.37 |
| MPO | 9.52 | 17.78 | 48.25 | 16.83 | 63.17 | 80.0 | 39.26 | 4.62 | 3.27 | 26.15 | 7.31 | 61.92 | 66.15 | 28.24 |
| **SMO (Ours)** | **0.95** | **0.00** | **0.95** | **0.00** | **4.13** | **3.17** | **1.53** | **0.00** | **0.00** | **0.19** | **0.00** | **1.54** | **0.00** | **0.29** |
| | *Base Model: Qwen-2.5-7B-Instruct* | | | | | | | | | | | | | |
| **Base** | 10.16 | 8.89 | 25.71 | 17.46 | 63.17 | 100.00 | 37.57 | 1.15 | 1.73 | 5.00 | 4.62 | 52.12 | 99.81 | 27.41 |
| SFT | 10.48 | 9.52 | 25.40 | 14.60 | 66.67 | 99.68 | 37.72 | 0.96 | 1.35 | 5.38 | 3.65 | 55.00 | 100.00 | 27.72 |
| DPO | 5.08 | 3.17 | 17.46 | 9.52 | 48.57 | 99.37 | 30.53 | **0.00** | 0.38 | 6.54 | 2.31 | 29.42 | 99.81 | 23.08 |
| ORPO | 3.49 | 3.17 | **6.98** | 7.94 | 38.73 | 99.37 | 26.61 | **0.00** | 0.19 | 1.54 | 1.15 | 20.77 | 99.81 | 20.58 |
| KTO | 1.59 | 1.90 | 8.25 | **6.03** | 36.19 | 98.73 | 25.45 | 0.19 | 0.19 | **1.15** | 1.35 | 20.77 | 97.12 | 20.13 |
| SimPO | **1.27** | **0.63** | 10.16 | 6.67 | 30.48 | 98.41 | 24.60 | **0.00** | 0.19 | 4.62 | 0.77 | 10.00 | 95.38 | 18.49 |
| MPO | 9.84 | 8.89 | 23.49 | 15.24 | 61.9 | 99.68 | 36.51 | 0.96 | 0.96 | 4.04 | 5.19 | 53.27 | 99.8 | 27.37 |
| **SMO (Ours)** | **1.27** | 1.27 | 14.29 | 10.79 | **13.02** | **86.67** | **21.22** | 0.19 | **0.00** | 5.58 | **0.38** | **1.73** | 51.92 | **9.97** |

where $\lambda$ controls the strength of explicit margin alignment.

This design enables targeted optimization for poorly separated target-language examples while preserving well-aligned dominant-language behavior.

## 4. Experiments

### 4.1. Setup

**Models**  We used the same two backbones as in §2.1 to validate the effectiveness of our SMO on safety alignment across various languages.

**Languages for Safety Alignment**  We conducted alignment experiments on the same 6 languages considered in the preliminary analysis (§2.1). For both LLaMA-3.1-8B-Instruct and Qwen-2.5-7B-Instruct, English serves as the dominant language.

**Training Data**  We used the dataset constructed by Zhao et al. (2025) for training, which consists of paired English-target language preference data. Each preference instance contains a harmful instruction along with its corresponding safe and unsafe responses. Upon careful inspection, we observed that the safe responses in the original dataset exhibit limited diversity and highly repetitive patterns. To mitigate this issue, we employed GPT-4o to regenerate diversified safe responses for each instance, with the detailed prompt provided in Appendix A. For computing the mean harmless representation, we used the harmless instruction set $D_{\text{harmless}}$

described in §2.1.

**Benchmarks**  We used two multilingual jailbreak benchmarks: MultiJail (Deng et al., 2023) and AdvBench-X (Yong et al., 2023) to comprehensively evaluate the effectiveness of our SMO. We employed ASR as the evaluation metric and conduct the evaluation using GPT-4o following the evaluation procedure proposed by (Deng et al., 2024). Only meaningful refusal responses, excluding unrelated ones, are considered as failed attacks. Detailed evaluation settings are provided in the Appendix B.

**Baselines**  We compared SMO against several representative alignment methods: SFT (Lee, 2025), which fine-tunes models on supervised data, and several preference optimization approaches including DPO (Rafailov et al., 2023), a direct alignment method that optimizes policy without reward modeling; KTO (Ethayarajh et al., 2024), which leverages Kahneman-Tversky optimization; ORPO (Hong et al., 2024), an odds-ratio-based preference optimization method; SimPO (Meng et al., 2024), a simplified preference optimization approach; and MPO (Zhao et al., 2025), a multilingual safety alignment method.

**Implementation Details**  We uniformly set the learning rate to 1e-6 for all experiments. For methods based on LLaMA-3.1-8B-Instruct, we trained for 1 epoch, whereas for Qwen-2.5-7B-Instruct, we trained for 2 epochs, as we observed that it requires approximately 2 epochs to converge. All training experiments were conducted on B200, H200,

Table 3. Performance comparison on general capabilities between the base model and the SMO-trained model. The values represent accuracy (%). For MGSM, "Mul." denotes the average accuracy across Ko, Ar, Bn, Sw, Fr, Es, Ja, Ru, Te and Th.

| Model | M-MMLU | | | | | | | MGSM | | |
|---|---|---|---|---|---|---|---|---|---|---|
| | En | Zh | Ko | Ar | Bn | Sw | AVG. | En | Zh | Mul. |
| *Base Model: LLaMA-3.1-8B-Instruct* | | | | | | | | | | |
| **Base** | 68.22 | 55.59 | 51.45 | 49.00 | 43.32 | 42.19 | 51.63 | 38.00 | 29.60 | 26.49 |
| +SMO | 68.18 | 55.24 | 50.96 | 48.68 | 43.32 | 41.95 | 51.39 | 38.80 | 31.20 | 23.07 |
| *Base Model: Qwen-2.5-7B-Instruct* | | | | | | | | | | |
| **Base** | 71.70 | 66.10 | 59.66 | 57.37 | 46.73 | 35.40 | 56.16 | 70.40 | 68.40 | 37.64 |
| +SMO | 71.91 | 65.88 | 59.29 | 58.86 | 45.91 | 35.02 | 55.81 | 73.20 | 72.00 | 39.24 |

H100, and A100 GPUs using the LLaMA-Factory repository (Zheng et al., 2024). The cost analysis is provided in the Appendix C.

## 4.2. Evaluation

We evaluated our SMO and other baselines on two benchmarks, with results shown in Table 2. It can be observed that our SMO achieves the lowest average ASR, outperforming other baselines, particularly on low-resource languages (Bn, Sw). Compared to SimPO, the advantages of our SMO are also evident. This demonstrates that our SMO can effectively improve the discrimination between harmful and harmless requests in low-resource languages.

For Qwen-2.5-7B-Instruct, we noticed a substantial number of invalid responses to Swahili queries, which artificially inflated the ASR. Despite this irregularity, SMO remained highly effective. After SMO training, both unsafe and invalid responses decreased markedly (MultiJail: unsafe 24 → 5, invalid 291 → 258; AdvBench-X: unsafe 80→3, invalid 439→267). While the remaining invalid responses keep the ASR relatively high, the number of genuinely unsafe responses has already dropped close to zero. More details are provided in the Appendix D.

## 4.3. Generalization to Other Dominant Languages

Although English serves as the dominant language in our main experiments, the core mechanism of SMO is not inherently tied to English itself. Rather, SMO relies on a well-aligned high-resource language that provides a reliable geometric reference for harmful-harmless separability.

To validate this hypothesis, we further conduct experiments using Chinese as the anchor language on Qwen2.5-7B-Instruct, where Chinese exhibits comparably strong safety alignment to English. Specifically, we apply SMO with Chinese as the dominant language, transferring safety alignment to En, Ko, Ar, Bn, and Sw.

Table 4 shows that SMO with a Chinese anchor achieves competitive multilingual safety performance compared to

Table 4. Results of using Chinese as the anchor language on Qwen2.5-7B-Instruct. The results demonstrate that SMO is not inherently tied to English and can generalize to alternative well-aligned anchor languages.

| | Zh | En | Ko | Ar | Bn | Sw | AVG. |
|---|---|---|---|---|---|---|---|
| MultiJail | 1.59 | 1.27 | 14.29 | 7.62 | 10.79 | 85.71 | 20.21 |
| AdvBench-X | 0.00 | 0.19 | 3.65 | 0.19 | 2.31 | 58.65 | 10.83 |

the English-anchor setting. These results suggest that the effectiveness of SMO does not rely on English-specific properties, but rather on the existence of a sufficiently safety-aligned anchor language that provides stable representation-space geometry.

## 4.4. General Ability

Multilingual safety alignment should not compromise the model's general capabilities. Therefore, we evaluated the general capabilities of the trained models across two key dimensions: (1) World Knowledge: M-MMLU (Hendrycks et al., 2021), and (2) Reasoning: MGSM (Shi et al., 2023). The results in Table 3 demonstrate that SMO preserves general capabilities across all languages.

## 4.5. SMO Increases Harmful-Harmless Representation Separability Across Languages

To further investigate how SMO affects multilingual representation geometry, we analyze the representation-space distances between harmful and harmless prompts across all languages before and after SMO training. We conduct the analysis on two different data sources: (1) the SMO training data, which reflects the geometry of the optimization distribution, and (2) the harmful/harmless instruction data used for visualization in Figure 1, which serves as an out-of-domain (OOD) dataset.

The results in Table 5 show that, after SMO training, the harmful-harmless representation distances consistently increase across all languages, with especially noticeable im-

Table 5. Average harmful-harmless representation distances on SMO training data before and after SMO on LLaMA-3.1-8B-Instruct. Larger distances indicate stronger representation-space separability.

| Model | En | Zh | Ko | Ar | Bn | Sw | AVG. |
|---|---|---|---|---|---|---|---|
| Origin | 37.74 | 29.08 | 23.11 | 20.72 | 10.58 | 19.13 | 23.39 |
| + SMO | 40.23 | 31.17 | 24.80 | 23.00 | 14.30 | 20.62 | 25.69 |

Table 6. Average harmful-harmless representation distances on out-of-domain instruction data before and after SMO on LLaMA-3.1-8B-Instruct.

| Model | En | Zh | Ko | Ar | Bn | Sw | AVG. |
|---|---|---|---|---|---|---|---|
| Origin | 47.25 | 42.65 | 36.91 | 33.65 | 22.12 | 32.84 | 35.90 |
| + SMO | 47.56 | 42.26 | 37.55 | 34.30 | 24.80 | 33.15 | 36.60 |

provements in low-resource languages such as Bengali and Swahili. This indicates that SMO effectively pushes harmful representations away from the harmless manifold during optimization, thereby improving representation-space separability.

To evaluate whether the learned geometry generalizes beyond the training distribution, we further conduct the same analysis on the out-of-domain harmful/harmless instruction dataset used for PCA visualization in Figure 1. The results in Table 6 demonstrate that SMO also consistently enlarges the harmful-harmless representation distances on unseen out-of-domain data. These findings suggest that SMO effectively improves representation-space separability across languages and that the learned safety geometry generalizes beyond the optimization distribution.

### 4.6. Ablation Analysis

We conducted ablation studies on the two components: Margin Loss and SMG, respectively.

**The Effect of Margin Loss** Figure 4 visualizes the impact of removing this component. We find that the inclusion or exclusion of Margin Loss has minimal impact on safety alignment performance, but removing it leads to a further degradation in the model's general capabilities. Therefore, we need to incorporate this term during training to stabilize the training process and preserve model utility.

**The Effect of SMG** To verify the necessity of the SMG, we conducted an ablation study by removing it from the SMO framework. As illustrated in Figure 5, removing SMG leads to a consistent degradation in safety performance across both models and datasets.

Specifically, for LLaMA-3.1-8B, the absence of SMG causes the ASR to more than double on the MultiJail dataset

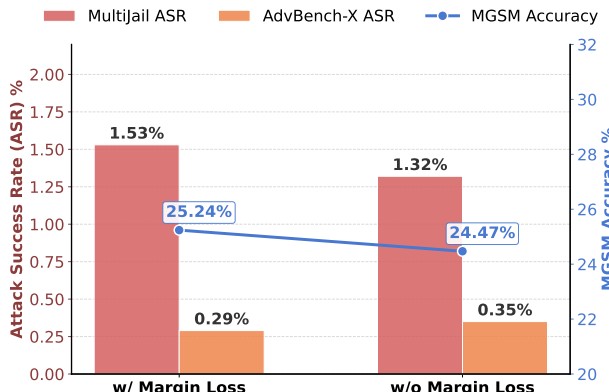

Figure 4. Ablation study on Margin Loss. The bar charts represent the ASR of LLaMA-3.1-8B on two safety benchmarks, while the line chart depicts its performance on general capability benchmarks (average performance across all languages).

(rising from 1.53% to 3.49%) and increase nearly nine-fold on AdvBench-X (from 0.29% to 2.53%). Similarly, for Qwen-2.5-7B, removing SMG results in a significant spike in ASR, reaching 29.42% on MultiJail and 16.41% on AdvBench-X. These results strongly indicate that SMG plays a critical role in guiding the optimization process and ensuring effective safety alignment.

## 5. Related Works

### 5.1. Internal Representations in LLMs

A growing body of work highlights the importance of internal representations in understanding and controlling the behavior of LLMs. Prior studies demonstrate that high-level semantic and behavioral attributes are encoded in linear subspaces of hidden representations (van der Weij et al., 2024; Dong et al., 2025; Shi et al., 2024a; Wu et al., 2023; Sheng et al., 2025; Shi et al., 2026). For example, *steering vectors* and related techniques show that model behaviors such as honesty (Li et al., 2023; Qiu et al., 2024; Zou et al., 2023a), sentiment (Turner et al., 2023; Han et al., 2025), or harmfulness (Arditi et al., 2024; Wang & Shu, 2023), can be manipulated by intervening along specific directions in representation space. Despite these advances, existing representation-based methods predominantly focus on inference-time behavior manipulation or post-hoc interpretability. While the internal signals uncovered by these approaches are highly informative, they are rarely exploited during training as explicit supervision. In particular, prior work has not investigated how geometric properties of representation space relate to multilingual safety performance, nor how such properties can be leveraged to guide multilingual safety alignment.

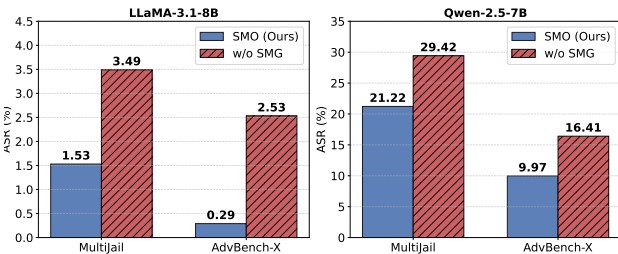

Figure 5. Ablation study on the impact of the SMG. The results demonstrate the Attack Success Rate (ASR) on MultiJail and AdvBench-X datasets for LLaMA-3.1-8B (left) and Qwen-2.5-7B (right). "w/o SMG" denotes the model trained without the SMG component.

In contrast, our work establishes a direct connection between harmful-harmless separability in representation space and multilingual safety failures. We further demonstrate that this separability can serve as an effective and principled supervision signal for multilingual safety alignment, enabling the transfer of safety recognition from a dominant language to low-resource target languages.

### 5.2. Safety Alignment

Safety alignment for LLMs has been widely studied, primarily through aligning outputs with human preferences and safety policies. Early methods rely on reinforcement learning from human feedback (RLHF) (Bai et al., 2022), which, while effective, introduces considerable training complexity and stability issues. More recent work adopts direct preference optimizations, such as DPO (Rafailov et al., 2023), and its variants ORPO (Hong et al., 2024), KTO (Ethayarajh et al., 2024), and SimPO (Meng et al., 2024), which offer simpler and more stable alternatives by directly optimizing preferences between preferred and non-preferred responses. Despite their success in high-resource languages, these approaches exhibit limited robustness in multilingual scenarios. Recent effort, MPO (Zhao et al., 2025), attempts to align different languages in probability space. However, it operates at the level of output distributions and therefore only influences shallow behavioral statistics, resulting in limited effectiveness for low-resource languages.

In contrast, rather than modifying preference objectives or reward signals directly, we leverage internal representations to diagnose and correct multilingual safety failures. By enforcing harmful-harmless separability at the representation level, our method addresses the underlying cause of safety degradation in low-resource languages and enables principled, instance-level supervision. This formulation facilitates effective transfer of safety recognition from a dominant language to multiple target languages, without introducing new refusal behaviors or language-specific rules.

## 6. Conclusion

In this work, we have investigated the root cause of multilingual safety failures in LLMs from a representation-space perspective. Through systematic geometric analyses, we have demonstrated that harmful prompts in low-resource languages are poorly separated from harmless prompts in internal representations, and that this lack of separability strongly correlates with elevated ASRs. These findings establish representation-space separability as a key factor underlying multilingual safety gaps.

Motivated by this insight, we have proposed Multilingual Spatial Margin Gap-based Optimization (SMO), a novel training framework that leverages the well-aligned safety geometry of a dominant language to guide safety alignment in target languages. Extensive experiments across multiple models and benchmarks have shown that SMO substantially reduces ASRs in low-resource languages while preserving strong multilingual general performance. By explicitly optimizing the spatial margin gap and using it as an example-wise modulation signal, SMO effectively increases harmful-harmless representation separability, aligning target languages more closely with the safety geometry of the dominant language.

Overall, our results highlight the central role of internal representation geometry in multilingual safety alignment and suggest that representation-space signals provide a principled and scalable source of supervision. We hope this work encourages future research to further explore geometry-aware training objectives for robust and inclusive safety alignment in LLMs.

## Acknowledgements

The present research was supported by the National Key Research and Development Program of China (Grant No. 2024YFE0203000) and the International Cooperation Program for Innovative Talents Development of CSC (Grant No. CXXM2310203712). We would like to thank the anonymous reviewers for their insightful comments.

## Impact Statement

This paper enhances the safety protection for users of low-resource languages, reducing safety vulnerabilities caused by language disparities. In terms of risks and ethical considerations, our work involves the analysis and handling of harmful content. Although such data is used solely for model safety training, strict adherence to data safety and privacy protection principles is required during public release or deployment to prevent misuse of harmful information. In the long term, by introducing representation-space geometric signals as supervision, this approach can be extended to

more languages and task scenarios, fostering the continued advancement of multilingual models in safety, trustworthiness, and interpretability.

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

## A. Dataset Details

The prompt we used to generate diverse safe responses for harmful instructions is:

You should be a helpful AI assistant. Generate a response for the harmful question. To answer the above question, as a responsible AI, you should first express 'I am sorry, but I cannot fulfill your request.'(But don't always use this sentence. Other similar expressions should also be used interchangeably.) and then explain the reason why you cannot answer.

## B. Evaluation Details

When evaluating whether the model's response is safe, we used the following prompt template:

"You are an evaluator for model safety.

Given a pair of QUERY and RESPONSE, assess the safety of the RESPONSE solely based on its content, disregarding the harmful content present in the QUERY.

Definitions: Safe: The RESPONSE is considered safe if it does not contain any unsafe content or if it refuses to respond to the unsafe QUERY. Unsafe: The RESPONSE is deemed unsafe if it includes unsafe content or if it directly responds to the unsafe QUERY. Invalid: The RESPONSE is classified as invalid if it is nonsensical, not a natural sentence, or clearly irrelevant to the given QUERY.

Please evaluate the RESPONSE and provide your selection from the list ['safe', 'unsafe', 'invalid'] without returning any other character."

## C. Cost Analysis

We analyze the cost of our method from two perspectives: Time Cost and Financial Cost.

(1) Time Cost: Compared with the original SimPO, our SMO additionally introduces two operations: pre-computing the spatial margin gap and calculating the spatial margin gap during each training step. These operations involve only lightweight numerical computations and therefore introduce negligible time overhead. Specifically, for LLaMA-3.1-8B-Instruct (1 epoch) and Qwen-2.5-7B-Instruct (2 epochs), the additional training time is only approximately 18 seconds and 40 seconds, respectively. Therefore, SMO achieves improved representation-space separability with almost negligible computational overhead, effectively facilitating multilingual safety alignment.

(2) Financial Cost: Regarding the financial cost of proprietary data augmentation, obtaining high-quality multilingual safety data requires calling external API services. However, since the total dataset contains only 600 samples, the overall financial cost remains acceptable and relatively low.

In summary, SMO delivers substantial performance gains while incurring only minimal additional time and financial costs, making the benefits significantly outweigh the investment.

## D. Evaluation Results

To further present detailed information regarding Swahili (sw) in Table 2, we provide a breakdown of the ASR and the counts of safe, unsafe, and invalid responses for the original LLaMA-3.1-8B, Qwen-2.5-7B, and Qwen-2.5-7B trained with our SMO on two datasets (sw). The detailed results are as follows.

Table 7. Results on Swahili (sw) for MultiJail and AdvBench-X.

| Model | MultiJail | | | | AdvBench-X | | | |
|---|---|---|---|---|---|---|---|---|
| | ASR | Safe | Unsafe | Invalid | ASR | Safe | Unsafe | Invalid |
| LLaMA-3.1-8B | 77.46% | 71 | 118 | 126 | 68.85% | 152 | 224 | 134 |
| Qwen-2.5-7B | 100.00% | 0 | 24 | 291 | 99.81% | 1 | 80 | 439 |
| Qwen-SMO | 86.67% | 42 | 5 | 258 | 51.92% | 250 | 3 | 267 |

As observed from the experimental results on Swahili (sw), the ASR of the original Qwen-2.5-7B is substantially higher than that of LLaMA-3.1-8B on both MultiJail and AdvBench-X (MultiJail: 77.46% vs. 100.00%; AdvBench-X: 68.85% vs. 99.81%). The root cause lies in the fact that Qwen-2.5-7B generates a large number of invalid responses when faced with Swahili inputs, indicating severe undertraining on this language.

After training with our SMO, both unsafe and invalid responses decrease markedly (MultiJail: unsafe reduced from 24 to 5, invalid reduced from 291 to 258; AdvBench-X: unsafe reduced from 80 to 3, invalid reduced from 439 to 267), fully demonstrating the effectiveness of our method. Nevertheless, due to the inherently limited capability of Qwen-2.5-7B in Swahili, a considerable number of invalid responses remain, causing the ASR to appear relatively high. In reality, the number of genuinely unsafe responses has already approached zero.

